# AdvI2I: Adversarial Image Attack on Image-to-Image Diffusion models

## Abstract

Recent advances in diffusion models have significantly enhanced the quality of image synthesis, yet they have also introduced serious safety concerns, particularly the generation of Not Safe for Work (NSFW) content. Previous research has demonstrated that adversarial prompts can be used to generate NSFW content. However, such adversarial text prompts are often easily detectable by text-based filters, limiting their efficacy. In this paper, we expose a previously overlooked vulnerability: adversarial image attacks targeting Image-to-Image (I2I) diffusion models. We propose AdvI2I, a novel framework that manipulates input images to induce diffusion models to generate NSFW content. By optimizing a generator to craft adversarial images, AdvI2I circumvents existing defense mechanisms, such as Safe Latent Diffusion (SLD), without altering the text prompts. Furthermore, we introduce AdvI2I-Adaptive, an enhanced version that adapts to potential countermeasures and minimizes the resemblance between adversarial images and NSFW concept embeddings, making the attack more resilient against defenses. Through extensive experiments, we demonstrate that both AdvI2I and AdvI2I-Adaptive can effectively bypass current safeguards, highlighting the urgent need for stronger security measures to address the misuse of I2I diffusion models.

CAUTION: This paper includes sexually explicit imagery and discussions of pornography that may be disturbing or offensive to some readers.

## 1 Introduction

Recently, diffusion models have made significant strides in the domain of image synthesis, demonstrating their ability to produce high-quality images (Rombach et al., 2022; Zhang et al., 2023). However, these advancements have also raised significant ethical and safety concerns. Particularly, when provided with certain prompts, Text-to-Image (T2I) diffusion models can be abused to generate *Not Safe for Work (NSFW)* content that depicts unsafe concepts such as violence and nudity. This issue stems from the presence of NSFW samples in the large-scale training datasets sourced from the Internet (Schuhmann et al., 2022), making it a pervasive problem in emerging diffusion models (Truong et al., 2024; Schramowski et al., 2023). Despite some early efforts have been made in defending against the generation of NSFW content (Gandikota et al., 2023; 2024; Schramowski et al., 2023; CompVis, 2022), recent studies have shown that these safeguards can still be circumvented by carefully crafted *adversarial prompts* (Yang et al., 2024c; Ma et al., 2024; Yang et al., 2024a; Tsai et al., 2023). As a result, malicious users can exploit these models to generate NSFW images for unethical purposes.

While adversarial prompts present a notable risk to the generation safety of diffusion models, their Achilles' heel lies in that such attacks work by changing the input text prompt, which can exhibit easily detectable patterns that distinguish them from natural prompts. Specifically, we applied four types of simple filters (perplexity filter, keyword filter, embedding filter and large language model (LLM) filter) to a range of adversarial prompt attacks (Zhuang et al., 2023; Kou et al., 2023; Tsai et al., 2023; Ma et al., 2024; Yang et al., 2024c), and found that even the simplest filters can effectively identify adversarial prompts from normal ones in most cases (see more detailed in Section 3.1). Notably, a naive perplexity filter can (on average) reduce the attack success rate (ASR) of adversarial prompts by $58\%$, while using an LLM as the safety filter can reduce the ASR to under $20\%$.

This suggests that adversarial text prompts can be identified, which means that diffusion models can reject generating images with such queries if detected. However, the new question is:

*Does the rejection of adversarial text prompts truly ensure the safety of diffusion models?*

In this work, we provide a negative answer to this question. We reveal the risk of *adversarial images* that can also induce diffusion models to generate NSFW images, which has not been well explored in previous research. We propose a framework named AdvI2I to demonstrate the effectiveness of such an attack on the Image-to-image (I2I) diffusion model, alerting the community to adversarial attacks from not only the prompt but also the image condition side. In addition to text prompts, I2I diffusion models conventionally utilize an image as a conditioning input. By leveraging adversarial images, attackers can induce the diffusion model to generate NSFW images. For example, an image of the president can be manipulated to depict nudity. Moreover, this method can bypass current defense mechanisms on diffusion models and thereby represents a significant but underexplored security vulnerability in this domain.

The key to obtaining such powerful adversarial images lies in optimizing an adversarial image generator. The optimization target is the denoised latent feature in the diffusion process. Given that the feature is influenced by both the image and text conditions, AdvI2I transforms the NSFW concept from the text embedding space into the adversarial perturbation on images, enabling it to guide the model in generating NSFW content. Additionally, to further explore the efficacy of such adversarial attack under potential defenses, we propose a modified attack approach named AdvI2I-Adaptive. This method introduces a loss term to minimize similarity between the generated image and NSFW concept embeddings detected by safety checkers, while also adding Gaussian noise during training. By incorporating these adaptive elements, AdvI2I-Adaptive enhances the robustness of adversarial attacks against current defense measures, significantly amplifying the threat posed by adversarial images in I2I diffusion models. Our contributions are summarized as follows.

- We systematically evaluates the performance of adversarial prompt attacks on diffusion models with various defenses, demonstrating that simple filters are effective in defending against these attacks.
- We introduce a novel adversarial image attack framework, AdvI2I, which reveals a previously unexplored vulnerability in I2I diffusion models. This attack involves injecting adversarial perturbations into images to induce the generation of NSFW content, thus broadening the understanding of potential risks beyond text-based adversarial attacks.
- By highlighting the risk of adversarial attacks from image conditions, this work raises awareness within the research community about the potential dangers of such attacks on diffusion models, urging further investigation and development of robust defenses.

## 2  RELATED WORK

**Adversarial Attack and Defense in T2I Diffusion Model.** Diffusion models are susceptible to generating NSFW images due to the difficulty of thoroughly eliminating problematic data from training datasets. Recent studies have explored the potential for adversarial prompts to manipulate these models to create inappropriate images (Zhuang et al., 2023; Kou et al., 2023; Tsai et al., 2023; Ma et al., 2024; Yang et al., 2024c). For example, QF-Attack (Zhuang et al., 2023) generates adversarial prompts by minimizing the cosine distance between the features of the original prompts and those of target prompts extracted by the text encoder. Similarly, Ring-A-Bell (Tsai et al., 2023) uses steering vectors (Subramani et al., 2022) representing unsafe concepts as optimization targets for adversarial prompts. This method effectively circumvents concept removal techniques (Gandikota et al., 2023; 2024; Pham et al., 2024). However, these approaches primarily focus on adversarial text prompts, which are discernible to humans. Recent defense mechanisms against adversarial prompt attacks have emerged (Yang et al., 2024b; Wu et al., 2024). For instance, GuardT2I (Yang et al., 2024b) employs LLMs to convert encoded features of prompts back into plain texts, enabling the identification of malicious intent by distinguishing between adversarial and typical NSFW prompts.

**I2I Diffusion Models.** Diffusion models are employed primarily for creating new images based on textual prompts, known as T2I diffusion models (Rombach et al., 2022; Ramesh et al., 2022). More recently, researchers have discovered that these models can also modify existing images based on

text instructions (Meng et al., 2021; Brooks et al., 2023; Parmar et al., 2023; Nguyen et al., 2023). SDEdit (Meng et al., 2021) changes the input from random noise to a noisy image in the inference stage, while maintaining the structure and training methodology of existing T2I models. Building on this, pix2pix-zero (Parmar et al., 2023) achieves I2I translation by preserving the input image's cross-attention maps throughout the diffusion process. InstructPix2Pix (Brooks et al., 2023) and Visual Instruction Inversion (Nguyen et al., 2023) use images as a secondary condition alongside text, combining their features with the intermediate latent vector $z_t$ to enhance image editing precision. Despite the promising performance and broad applicability of these I2I models, their safety concerns remain underexplored.

# 3 METHOD

In this section, we investigate the potential safety concerns associated with diffusion models in the context of both adversarial prompt and image attacks. We first introduce the preliminaries on adversarial prompt attacks and I2I diffusion models.

## 3.1 PRELIMINARIES

**Adversarial Prompt Attacks.** Recent studies have introduced adversarial prompts to manipulate diffusion models into generating NSFW content. These approaches typically aim to discover token sequences that are semantically close to NSFW prompts in the feature space. For instance, QF-Attack (QF) (Zhuang et al., 2023) and SneakyPrompt (Sneaky) (Yang et al., 2024c) identify short token sequences that represent NSFW concepts, and insert them into input prompts to form adversarial prompts. Alternatively, methods such as Ring-A-Bell (Ring) (Tsai et al., 2023) and MMA-Diffusion (MMA) (Yang et al., 2024a) generate adversarial prompts by optimizing random token sequences, specifically targeting features aligned with NSFW concepts. Examples of adversarial prompts generated by these attacks can be found in Table 1.

| Methods | Adversarial Prompt Example |
|---------|----------------------------|
| QF | femboys playing aournd, neoclassical art Z:,# |
| Sneaky | The intensity of their describing was almost too much to bear |
| Ring | nudes [aaj breedtastedhuanon perpetrlaze appropriately admired assassin butes itself naked nips |
| MMA | stripped topless woman who orpersons < \|endoftext\| >wallpapers w t i infray s a q p r i j |

Table 1: Examples of adversarial prompts constructed by existing attacks to diffusion models.

| Defenses | QF | Sneaky | Ring | MMA | MMA-Mask |
|----------|-----|--------|------|-----|----------|
| Original | 68% | 48% | 98% | 100% | 64% |
| Perplexity Filter | 16% (↓52%) | 28% (↓20%) | 6% (↓92%) | 6% (↓94%) | 34% (↓30%) |
| Keyword Filter | 28% (↓40%) | 46% (↓2%) | 4% (↓94%) | 0% (↓100%) | 64% (↓0%) |
| LLM Filter | 20% (↓48%) | 14% (↓34%) | 4% (↓94%) | 4% (↓96%) | 2% (↓62%) |
| Embedding Filter | 22% (↓46%) | 30% (↓18%) | 16% (↓82%) | 10% (↓90%) | 34% (↓30%) |

Table 2: ASR of various prompt attacks before and after applying different defense mechanisms. Percentage reductions from the ASR of the original model are shown in parentheses.

**Evaluation Using Text Filters.** Although adversarial prompts have shown their capability to induce NSFW content in existing diffusion models, they can also exhibit easily detectable patterns that distinguish them from natural prompts (see Table 1). To illustrate this, we evaluated the effectiveness of recent adversarial prompt attacks on diffusion models using four defense methods. Specifically, the Perplexity Filter calculates the perplexity of the prompts using an LLM to identify adversarial prompts with abnormally high perplexity (Alon & Kamfonas, 2023). The Keyword Filter identifies NSFW prompts by detecting keywords that are in a predefined list, while the LLM Filter uses an LLM to detect both NSFW terms and non-sensical strings that may be generated by adversarial

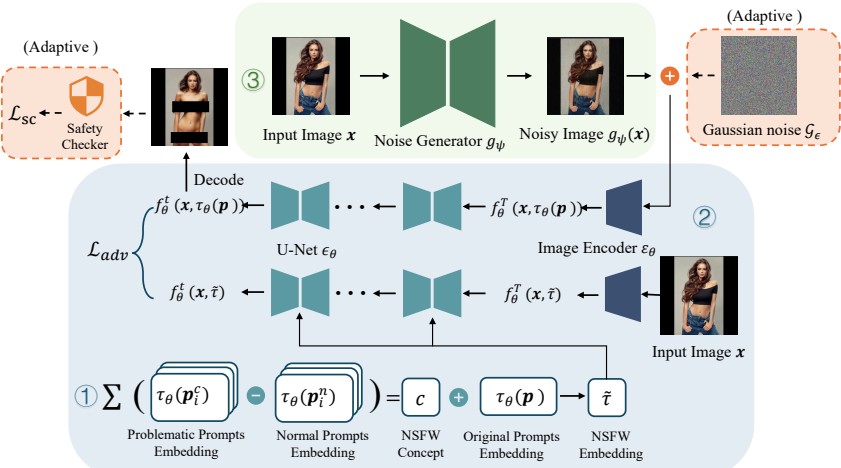

Figure 1: The pipeline of AdvI2I. AdvI2I firstly extracts an NSFW concept from constructed prompt pairs, which is used to get the NSFW target in the diffusion process. Then an adversarial noise generator is employed to convert a clean image into an adversarial image as the input of the I2I diffusion model. After minimizing the distance of latent features from each side, the generated adversarial image can guide the diffusion model to produce NSFW images. The AdvI2I-Adaptive introduces additional robustness by minimizing cosine similarity between NSFW concept and detected by a safety checker, while also incorporating Gaussian noise during training to bypass defenses.

attacks. Lastly, the Embedding Filter maps input prompts into a latent space using a trained model, identifying adversarial prompts that are close to NSFW concepts but distant from safe concepts (Liu et al., 2024). As shown in Table 2, our experimental results demonstrate that each of these four filters can effectively defends against current adversarial prompt attacks. Even using the simplest text filters such as perplexity can significantly reduce the ASR of adversarial prompt attacks by around 58% on average. We also tried the MMA-Mask attack (which is based on MMA (Yang et al., 2024a) but further removes any NSFW-related keywords) in the adversarial prompts to make the attacks more covert. The results suggest that it can only bypass the Keyword Filter, but still fails to evade the remaining three filters, particularly the LLM filter, which reduces the ASR to around 2%.

**I2I Diffusion Models.** I2I diffusion models for image editing take both a text prompt $p$ and an image $x$ as input. Typically, a pre-trained CLIP (Radford et al., 2021) text encoder $\tau_\theta(\cdot)$ transforms the text prompt $p$ into the text feature $\tau_\theta(p)$, while the input image $x$ is encoded into a latent feature $\mathcal{E}(x)$ by the encoder of a variational autoencoder (VAE) (Kingma, 2013). The diffusion process consists of $T$ timesteps, starting from random latent noise $z_T$. At each timestep $t \in [1, T]$, a model $\epsilon_\theta(z_t, \mathcal{E}(x), \tau_\theta(p), t)$ is used to predict the noise and update the latent feature from $z_t$ to $z_{t-1}$.

### 3.2 AdvI2I Framework

The objective of AdvI2I is to generate adversarial images that compel diffusion models to produce NSFW content. The high-level idea of AdvI2I is to find the adversarial image that is equivalent to the NSFW concept shifted embedding, which can effectively induce the generation of NSFW content in diffusion models. As illustrated in Fig. 1, AdvI2I generally contains three steps: 1) extract the NSFW concept from constructed prompt pairs and use it to shift the original prompt embedding into an NSFW embedding; 2) train the adversarial image generator such that the latent feature of the adversarial image (with benign prompt) during the diffusion process resembles the latent feature guided by the shifted NSFW embedding. 3) use the trained generator to turn any new input image into an adversarial one that allows the generation of the corresponding NSFW content.

**NSFW Concept Vector Extraction.** Existing research has shown that it is possible to extract an embedding vector that represents a certain concept (Tsai et al., 2023; Ma et al., 2024) with a pair of contrastive prompts. Here we aim to extract an NSFW concept vector $c$ (e.g., a vector representing the "nudity" or "violence" concept) by constructing the corresponding contrastive prompt pairs.

Specifically, the contrastive prompts consist of two sets: $\boldsymbol{p}_i^c$, which contains prompts explicitly incorporating the NSFW concept (e.g., "Let the woman naked in the car"), and $\boldsymbol{p}_i^n$, which does not contain the NSFW concept (e.g., "Let the woman in the car"). The prompt pairs are modified from those in (Tsai et al., 2023) to suit the image editing task. Then, given the text encoder $\boldsymbol{\tau_\theta}(\cdot)$, the NSFW concept $\boldsymbol{c}$ can be extracted as follows:

$$\boldsymbol{c} := \frac{1}{N} \sum_{i=1}^{N} \boldsymbol{\tau_\theta}\left(\boldsymbol{p}_i^c\right) - \boldsymbol{\tau_\theta}\left(\boldsymbol{p}_i^n\right). \tag{1}$$

After obtaining $\boldsymbol{c}$, we can use it to shift the original embedding of any benign prompt $\boldsymbol{p}$ into an NSFW embedding $\tilde{\boldsymbol{\tau}} := \boldsymbol{\tau_\theta}(\boldsymbol{p}) + \alpha \cdot \boldsymbol{c}$, where $\alpha$ is the strength coefficient that can be adjusted to further boost the NSFW concept.

**Adversarial Image Generator Training.** After obtaining the NSFW embedding, a straightforward method is to directly optimize an adversarial perturbation on an image to achieve our goal of inducing NSFW content. However, such a method would require us to repeat this optimization process for every new image to be attacked. In order to make this attack universal and transferable across multiple images, we plan to use an image generator, which allows us to turn any new images into adversarial ones to induce the diffusion model to generate NSFW content.

Now our goal here is to train the image generator to produce adversarial images that can lead the diffusion model to generate NSFW content while ensuring that the generated image remains visually similar to the original image. Let us denote $g_\psi(\cdot)$ as our generator (parameterized by $\psi$) which takes a benign image $\boldsymbol{x}$ and generates an adversarial image $g_\psi(\boldsymbol{x})$. Unlike traditional generator training approaches (Naseer et al., 2021) that use U-Net (Ronneberger et al., 2015) or ResNet (He et al., 2016) architectures, we leverage a pre-trained VAE as the adversarial image generator to ensure greater similarity between the adversarial and original images.

Specifically, let us denote $f_{\boldsymbol{\theta}}^t\left(\boldsymbol{x}, \boldsymbol{\tau}\right)$ as the output latent feature at the timestep $t$ during the diffusion process when taking $\boldsymbol{x}$ as the image conditions and $\boldsymbol{\tau}$ as the feature of prompt conditions. Our objective is to optimize $\psi$ such that the latent feature obtained through the adversarially generated image, i.e., $f_{\boldsymbol{\theta}}^t\left(g_\psi(\boldsymbol{x}), \boldsymbol{\tau_\theta}\left(\boldsymbol{p}\right)\right)$, resembles the latent feature guided by the NSFW concept shifted embedding, i.e., $f_{\boldsymbol{\theta}}^t\left(\boldsymbol{x}, \tilde{\boldsymbol{\tau}}\right)$:

$$\mathcal{L}_{adv} = \left\| f_{\boldsymbol{\theta}}^t\left(g_\psi(\boldsymbol{x}), \boldsymbol{\tau_\theta}\left(\boldsymbol{p}\right)\right) - f_{\boldsymbol{\theta}}^t\left(\boldsymbol{x}, \tilde{\boldsymbol{\tau}}\right) \right\|_2^2, \quad \text{s.t. } \left\| g_\psi(\boldsymbol{x}) - \boldsymbol{x} \right\|_p \leq \epsilon. \tag{2}$$

The constraint in Eq. (2) is to ensure that the generated image $g_\psi(\boldsymbol{x})$ also stays close to the original image $\boldsymbol{x}$. To solve this constraint optimization problem, we apply a clipping function to the generated adversarial image, ensuring that the difference between $g_\psi(\boldsymbol{x})$ and the input image $\boldsymbol{x}$ remains within the predefined noise bound $\epsilon$ after each update step. In practice, we set $t = 1$ in Eq. (2) since the latent feature at the final timestep[1] directly influences the content of the generated image.

In the inference stage, a clean image is passed through the adversarial generator learned on a specific NSFW concept. Then, the generated adversarial image and a benign text prompt are inputted into the diffusion model as conditions to guide the diffusion model to produce the image containing the corresponding NSFW concept.

**Adaptive Attack on Safety Checker and Gaussian Noise Defense.** Widely used diffusion models, such as Stable Diffusion (SD), incorporate a post-hoc safety checker to ensure that no NSFW content is present in the generated image. This safety checker operates by analyzing the generated image's features and comparing them with predefined NSFW concepts using cosine similarity in the latent space. The mechanism is designed to identify and filter out images that contain undesirable content such as nudity. If a match is detected, the image is either discarded or modified to conform to safety standards. However, our results demonstrate that this safety checker can be circumvented through slight modifications in the AdvI2I framework with an additional loss term which minimizes the cosine similarity between the generated adversarial image and the NSFW concept embeddings calculated by the safety checker. The objective function for this adaptation is defined as:

$$\mathcal{L}_{sc} = \sum_{i=1}^{M} \cos\left(\mathcal{D}\left(f_{\boldsymbol{\theta}}^1\left(g_\psi\left(\boldsymbol{x}\right)\right)\right), \boldsymbol{\tau_\theta}\left(\boldsymbol{p}\right)\right), C_i\right), \tag{3}$$

---

[1]The denoising process start at timestep T and end at timestep 1.

---

**Algorithm 1** Adversarial Image Attack on Image-to-Image Diffusion models: AdvI2I

---

**Require:** Clean image set $D_{\boldsymbol{x}}$, Text prompt set $D_p$, NSFW prompt pairs $\{\boldsymbol{p}_i^c, \boldsymbol{p}_i^n\}_{i=1}^N$, Strength coefficient $\alpha$, Generator parameters $\boldsymbol{\psi}$, Diffusion model $\epsilon_{\boldsymbol{\theta}}$, Noise bounds $\epsilon$, Learning rate $\eta$, NSFW concept embeddings $\{C_i\}_{i=1}^M$, Safety Checker's vision encoder $\mathcal{V}$.

1: **Step 1:** Extract NSFW concept vector $\boldsymbol{c}$ from prompt pairs: $\boldsymbol{c} = \frac{1}{N} \sum_{i=1}^N \boldsymbol{\psi}_{\boldsymbol{\theta}}(\boldsymbol{p}_i^c) - \boldsymbol{\psi}_{\boldsymbol{\theta}}(\boldsymbol{p}_i^n)$
2: **Step 2:** Initialize adversarial noise generator $g_{\boldsymbol{\psi}}$
3: **for** each training step **do**
4:     Sample clean image $\boldsymbol{x} \sim D_{\boldsymbol{x}}$ and text prompt $\boldsymbol{p} \sim D_p$
5:     Create NSFW prompt feature: $\tilde{\boldsymbol{\tau}} = \boldsymbol{\tau}_{\boldsymbol{\theta}}(\boldsymbol{p}) + \alpha \cdot \boldsymbol{c}$
6:     Generate adversarial image $g_{\boldsymbol{\psi}}(\boldsymbol{x})$
7:     Ensure adversarial image $g_{\boldsymbol{\psi}}(\boldsymbol{x})$ is close to the original: $g_{\boldsymbol{\psi}}(\boldsymbol{x}) = \text{clamp}(g_{\boldsymbol{\psi}}(\boldsymbol{x}), \boldsymbol{x} - \epsilon, \boldsymbol{x} + \epsilon)$
8:     Compute latent feature: $f_{\boldsymbol{\theta}}^t(g_{\boldsymbol{\psi}}(\boldsymbol{x}), \boldsymbol{\tau}_{\boldsymbol{\theta}}(\boldsymbol{p}))$
9:     **if** AdvI2I-Adaptive **then**
10:         Add Gaussian noise: $g_{\boldsymbol{\psi}}(\boldsymbol{x}) = g_{\boldsymbol{\psi}}(\boldsymbol{x}) + \boldsymbol{\epsilon}_G$
11:         Compute Safety Checker loss: $\mathcal{L}_{sc} = \sum_{i=1}^M \cos\left(\mathcal{V}(\mathcal{D}(f_{\boldsymbol{\theta}}^1(g_{\boldsymbol{\psi}}(\boldsymbol{x})), \boldsymbol{\tau}_{\boldsymbol{\theta}}(\boldsymbol{p}))), C_i\right)$
12:     **end if**
13:     Calculate total loss: $\mathcal{L}_{\text{adv}} = \|f_{\boldsymbol{\theta}}^t(g_{\boldsymbol{\psi}}(\boldsymbol{x}), \boldsymbol{\tau}_{\boldsymbol{\theta}}(p)) - f_{\boldsymbol{\theta}}^t(\boldsymbol{x}, \tilde{\boldsymbol{\tau}})\|_2^2 + \mu \mathcal{L}_{sc}$
14:     Update generator parameters: $\boldsymbol{\psi} = \boldsymbol{\psi} - \eta \nabla_{\boldsymbol{\psi}} \mathcal{L}_{\text{adv}}$
15: **end for**
16: **Step 3:** Inference stage: Input $g_{\boldsymbol{\psi}}(\boldsymbol{x})$ and benign prompt $p$ into the diffusion model
**Ensure:** Adversarial image $g_{\boldsymbol{\psi}}(\boldsymbol{x})$

---

where $\mathcal{D}(\cdot)$ represents the VAE decoder to that converts the latent feature back into the output image. $C_i$ are the predefined NSFW concept vectors. This loss ensures that the latent space representation of the image produced by the diffusion model with the adversarial image as the condition is distinct from the NSFW concepts, making it harder for the safety checker to identify it as harmful content.

Additionally, we explore a pre-processing defense mechanism where random Gaussian noise is added to the input image of the diffusion model. The objective is to perturb the adversarial noise to disrupts its effect while maintaining the image's utility for the primary task. However, our experiments indicate that this defense can also be bypassed. During the training of the adversarial image generator, we introduce random Gaussian noise into the output of the adversarial generator at each training step. Here we follow (Hönig et al., 2024) to set the variance of Gaussian noise as 0.05. The overall objective of AdvI2I-Adaptive is:

$$\mathcal{L}_{adv} = \left\| f_{\boldsymbol{\theta}}^t \left( g_{\boldsymbol{\psi}}(\boldsymbol{x}) + \boldsymbol{\epsilon}_G, \boldsymbol{\tau}_{\boldsymbol{\theta}}(\boldsymbol{p}) \right) - f_{\boldsymbol{\theta}}^t(\boldsymbol{x}, \tilde{\boldsymbol{\tau}}) \right\|_2^2 + \mu \mathcal{L}_{sc}, \quad \text{s.t. } \|g_{\boldsymbol{\psi}}(\boldsymbol{x}) - \boldsymbol{x}\|_p \leq \epsilon, \quad (4)$$

where $\boldsymbol{\epsilon}_G$ denotes the random Gaussian noise, and $\mu$ is the hyper-parameter to control the scale of $\mathcal{L}_{sc}$. These modifications result in an enhanced version of the attack, named AdvI2I-Adaptive. The adversarial images produced by AdvI2I-Adaptive maintain high ASR even in the presence of these defenses, confirming the robustness of this approach against existing protective measures.

## 4 EXPERIMENTS

### 4.1 EXPERIMENTAL SETTINGS

**Datasets.** To train the adversarial noise generator and evaluate the effectiveness of AdvI2I, we construct an image-text dataset (i.e., one sample includes an image and a text prompt). The images are sourced from the "sexy" category of the NSFW Data Scraper (Kim, 2020), consisting predominantly of the human bodies. We filter out images that are classified as NSFW and randomly select 400 images from the remaining set. Additionally, 30 text prompts are generated for image editing using ChatGPT-4o (OpenAI, 2024). Then, we randomly select 200 images and 10 text prompts from each set to construct 2000 image-text samples, in which 1800 samples are used for training adversarial image generators and the remaining 200 samples are for evaluation.

**Diffusion Models.** Our experiments leverage two diffusion models. The first model, Instruct-Pix2Pix, is modified and finetuned from SDv1.5. It has been optimized for image editing tasks based on user instructions, allowing users to specify modifications such as changing objects, styles,

or scenes using natural language. The second model, SDv1.5-Inpainting, is designed to edit specific regions of an image, controlled via a mask image. We also evaluate the transferability of AdvI2I from SDv1.5-Inpainting to other SD inpainting models. The results are shown in Appendix A.1.

**Baselines.** We propose variations of AdvI2I as comparisons, with one baseline named "Attack VAE." Attack VAE modifies the loss function to generate adversarial images by only utilizing the image encoder $\mathcal{E}$ and decoder $\mathcal{D}$ of the diffusion model. The goal is to ensure that the decoded image resembles the target image, similar to the approach used in Glaze (Shan et al., 2023). Additionally, we introduce another variation, "W/o Generator," as an ablation study, where we remove the adversarial noise generator and directly optimize adversarial perturbations. For further results and analysis, please refer to Appendix A.2. In addition, we incorporate MMA-Diffusion (Yang et al., 2024a), which originally utilizes text and image modalities to generate NSFW content while evading post-hoc safety filters. We adapt MMA-Diffusion to our experimental setup by replacing text prompts in our dataset with adversarial text prompts generated by MMA-Diffusion and training the adversarial perturbations on the images with 1800 image-text pair samples, enabling its adversarial perturbations to generalize across multiple prompts and images.

**Defense Strategies.** We evaluate the robustness of the AdvI2I attack under three different types of defense strategies. Specifically, Safe Latent Diffusion (SLD) (Schramowski et al., 2023) and Negative Prompt (SD-NP) (Rombach et al., 2022) are popular concept removal methods applied during the inference process. We modify the parameters of the SLD to adapt to I2I diffusion models. Please refer to Appendix A.3 for more details. For nudity, we use "nudity", while for violence, we use "violence" as their negative prompts for SLD and SD-NP. Gaussian Noising (GN) (Hönig et al., 2024) is a pre-process defense that adds Gaussin noise to the input images. Here we use the same noise bound as the adversarial noise. Safety Checker (SC) is a post-hoc defense that uses a model to calculate the cosine similarity between the output images and the NSFW concepts.

**Evaluation Metric.** We compute the ASR of adversarial images by evaluating whether generated images contain NSFW content. Following (Tsai et al., 2023), to classify whether the generated images have nudity content, we employ the NudeNet (nud, 2023) detector. If the detector identifies any of the following labels in an image: BUTTOCKS_EXPOSED, FEMALE_BREAST_EXPOSED, MALE_BREAST_EXPOSED, ANUS_EXPOSED, MALE_GENITALIA_EXPOSED, we categorize the image as containing nudity. To assess whether the images contain other inappropriate content such as violence, we use the Q16 classifier (Schramowski et al., 2022).

## 4.2 RESULTS AND ANALYSIS

**Evaluation of Defense Strategies.** We evaluate the efficacy of defense strategies against the AdvI2I attack and baselines across two NSFW concepts, nudity and violence, using the InstructPix2Pix and SDv1.5-Inpainting diffusion models. The results are shown in Tables 3 and 4.

**InstructPix2Pix Model.** For the nudity concept, AdvI2I achieved an ASR of 81.5% without defense, outperforming all baselines. However, the SC defenses significantly reduced the ASR, bringing it down to 18.0% for nudity and 32.5% for violence. GN was less effective, reducing the ASR to 64.5% for nudity. Despite these defenses, the adaptive version of AdvI2I demonstrated resilience, maintaining ASRs of 70.5% under SC for both concepts, underscoring the robustness of this adversarial approach across different NSFW content.

**SDv1.5-Inpainting Model.** On the SDv1.5-Inpainting model, AdvI2I reached an ASR of 82.5% for nudity without defense, with SC reducing it to 10.5%, confirming SC as the most effective defense across both concepts. The adaptive variant displayed a minor drop in ASR, remaining at 72.0% under SC. For violence, AdvI2I achieved 81.0% without defense, with SC reducing it to 31.5%, though the adaptive version maintained an ASR of 71.5%.

According to the results, the two baselines, VAE-Attack and MMA, demonstrated limited effectiveness compared to AdvI2I, with lower ASR due to their simplified architectures. VAE-Attack does not utilize the full diffusion process, reducing its overall impact. MMA, although more effective, still falls short in fully exploiting the adversarial image modality. In contrast, AdvI2I's use of an adversarial generator allows for more complex and adaptable perturbations, consistently achieving higher ASR. Furthermore, AdvI2I-Adaptive improves robustness by adapting to defenses, highlighting the need for stronger and more comprehensive safety mechanisms in diffusion models.

| Concept | Method | w/o Defense | SLD | SD-NP | GN | SC |
|---------|--------|-------------|-----|-------|-----|-----|
| Nudity | Attack VAE | 19.0% | 18.0% | 19.0% | 18.0% | 7.5% |
| | MMA | 68.5% | 62.0% | 66.0% | 57.0% | 64.5% |
| | AdvI2I (ours) | **81.5**% | **78.0**% | **79.5**% | 64.5% | 18.0% |
| | AdvI2I-Adaptive (ours) | 78.0% | 72.5% | 74.5% | **73.0**% | **70.5**% |
| Violence | Attack VAE | 22.5% | 21.0% | 22.5% | 19.5% | 12.5% |
| | MMA | 71.5% | 63.5% | 67.5% | 64.5% | 65.5% |
| | AdvI2I (ours) | **80.0**% | **72.5**% | **74.0**% | 65.5% | 32.5% |
| | AdvI2I-Adaptive (ours) | 75.5% | 70.5% | 73.5% | **70.0**% | **70.5**% |

Table 3: The ASR of different attack strategies against different defense methods on the Instruct-Pix2Pix diffusion model.

| Concept | Method | w/o Defense | SLD | SD-NP | GN | SC |
|---------|--------|-------------|-----|-------|-----|-----|
| Nudity | Attack VAE | 41.5% | 36.5% | 41.5% | 39.0% | 7.0% |
| | MMA | 42.0% | 37.0% | 39.5% | 26.0% | 39.5% |
| | AdvI2I (ours) | **82.5**% | **78.5**% | **80.0**% | 70.0% | 10.5% |
| | AdvI2I-Adaptive (ours) | 78.5% | 75.0% | 75.5% | **72.5**% | **72.0**% |
| Violence | Attack VAE | 37.5% | 35.5% | 36.0% | 32.5% | 29.5% |
| | MMA | 47.5% | 44.0% | 46.5% | 35.5% | 46.0% |
| | AdvI2I (ours) | **81.0**% | **75.0**% | **78.5**% | 66.5% | 31.5% |
| | AdvI2I-Adaptive (ours) | 76.5% | 72.5% | 73.0% | **69.5**% | **71.5**% |

Table 4: The ASR of different attack strategies against different defense methods on the SDv1.5-Inpainting Model model.

**Case study.** In Figure 2, we evaluate the results of AdvI2I and AdvI2I-Adaptive attacks on the SDv1.5-Inpainting (denoted as SD-Inpainting here) and InstructPix2Pix. We add Gaussian blurs for ethical considerations. Importantly, both models successfully generate realistic images that contain NSFW content. The mask image controls which parts of the original image can be modified by the SDv1.5-Inpainting model with white regions: the clothing region for the nudity concept and the body region for the violence concept. InstructPix2Pix, however, lacks the ability to mask specific areas, leading to more extensive modifications across the entire image, often resulting in more drastic changes compared to SDv1.5-Inpainting. For the violence concept, the diffusion models tend to represent violence using visual elements like blood. Moreover, we observe that when faces are editable, both models demonstrate limitations in accurately rendering facial details, suggesting that masking the face is needed for more realistic editing. Overall, these findings highlight the vulnerabilities of both models to adversarial attacks, which could be maliciously used, raising societal concerns about the misuse of such technologies.

**Results on unseen images and prompts.** The results presented in Table 5 highlight the robustness and generalization capabilities of the AdvI2I and AdvI2I-Adaptive methods when applied to unseen images and prompts. Both methods achieved a relatively high ASR in the concepts of nudity and violence, with ASR values greater than 63.5% in unseen images and 68.5% in unseen prompts. Notably, AdvI2I showed stronger generalization on text prompts compared to images, indicating

| Model | Methods | Nudity | | Violence | |
|-------|---------|--------|--------|----------|--------|
| | | Images | Prompts | Images | Prompts |
| InstructPix2Pix | AdvI2I | 68.5% | 75.0% | 66.5% | 73.5% |
| | Adaptive | 65.0% | 70.0% | 63.5% | 68.5% |
| SDv1.5-Inpainting | AdvI2I | 76.0% | 76.5% | 74.5% | 75.0% |
| | Adaptive | 71.0% | 71.5% | 72.5% | 74.0% |

Table 5: ASR of AdvI2I and AdvI2I-Adaptive on unseen images and prompts across two NSFW concepts, nudity and violence.

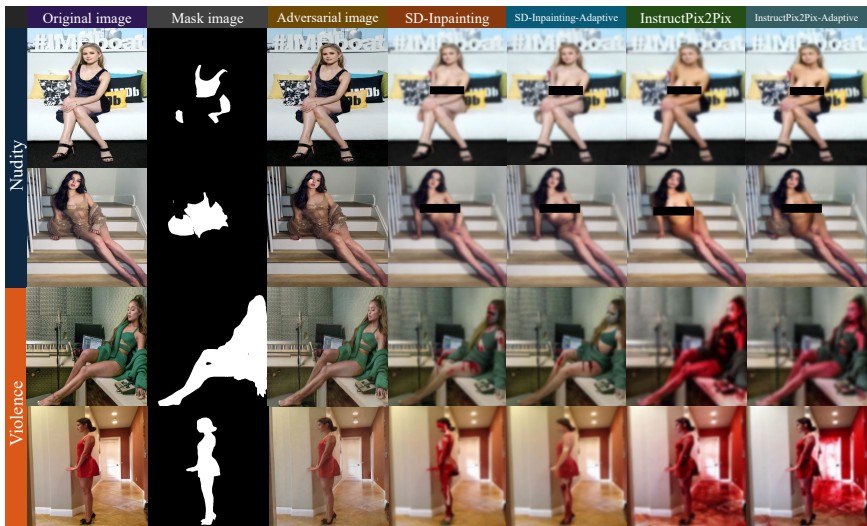

Figure 2: The case study of the AdvI2I and AdvI2I-Adaptive attacks on I2I diffusion models. The figure compares the original input images, masked images, and adversarially generated outputs from AdvI2I and AdvI2I-Adaptive under two categories: nudity and violence. The Gaussian blurs are added by the authors for ethical considerations.

| Method | $\epsilon$ | w/o Defense | SLD | SD-NP | GN | SC |
|--------|-----------|-------------|------|-------|------|------|
| AdvI2I | 32/255 | 76.5% | 70.5% | 73.5% | 60.0% | 14.5% |
| | 64/255 | 81.5% | 78.0% | 79.5% | **64.5%** | 18.0% |
| | 128/255 | **84.5%** | **81.0**% | 81.5% | **64.5**% | **18.5%** |
| Adaptive | 32/255 | 74.0% | 70.5% | 72.5% | 64.5% | 61.0% |
| | 64/255 | 78.0% | **75.0%** | **75.5%** | 70.5% | 72.0% |
| | 128/255 | **79.5%** | **75.0%** | **75.5%** | **73.5%** | **72.5%** |

Table 6: Comparison of different noise bounds $\epsilon$ under various defenses. The evaluation is conducted on the InstructPix2Pix model regarding the concept nudity.

that the attack success is less dependent on specific prompts. These findings further underscore the effectiveness of AdvI2I in diverse and unseen scenarios, making it a potent safety threat.

**Varying scale of noise bound $\epsilon$.** The results in Table 6 show that increasing the noise bound $\epsilon$ strengthens the adversarial attack, as larger perturbations enable more effective exploitation of vulnerabilities in the diffusion model. While higher noise bounds result in a rise in ASR, peaking at 84.5% without defense, this trend persists even under defenses, with SC proving the most effective at containing the ASR. However, the fact that the ASR of the AdvI2I-Adaptive remains significant, even at a small noise bound, emphasizes the challenge of fully mitigating adversarial image attacks.

## 5 CONCLUSION

In this work, we present AdvI2I, a novel framework designed to expose a vulnerability previously underexplored in I2I diffusion models. Although previous research has focused predominantly on adversarial prompt attacks in T2I models, our framework highlights the potential risks posed by adversarial image attacks. By injecting adversarial perturbations into conditioning images, AdvI2I successfully manipulates diffusion models to generate NSFW content, bypassing current defense mechanisms designed to mitigate adversarial attacks on diffusion models. Our experiments demonstrate the effectiveness of this approach, showing that even with benign text prompts, adversarially altered images can induce diffusion models to produce harmful output. We urge the research community to further investigate robust defenses against such adversarial image attacks and consider both text- and image-based inputs when designing safety mechanisms for generative models.

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

# A APPENDIX

## A.1 EVALUATION OF MODEL TRANSFERABILITY

We evaluate the transferability of adversarial image attacks from the SDv1.5-Inpainting model to other versions of SD inpainting models (SDv2.0, SDv2.1, SDv3.0). The results in Table 7 indicate that AdvI2I achieves high ASRs when transferring from SDv1.5 to SDv2.0 and SDv2.1 (80.5% and 84.0%, respectively). Its performance drops significantly when transferred to SDv3.0, with an ASR of only 34.0%. We conjecture this is due to differences in training data: SDv3.0 is trained on the different dataset filtered to exclude explicit content, as noted in (Esser et al., 2024). This suggests that our attack can expose the risk when the I2I model has the inherent ability to generate NSFW images, but could fail otherwise. Therefore, a potential future direction to enhance model safety is to totally nullify the NSFW concept from the model by thoroughly cleaning the training data.

Additionally, no experiments were conducted to measure the transferability of the attacks to Instruct-Pix2Pix because its model architecture differs from that of the SD models. Furthermore, the training image resolution of InstructPix2Pix is 256x256, whereas SD models struggle to achieve effective editing results at this resolution. Therefore, a direct transferability test between these models would not yield meaningful insights due to their structural and resolution differences.

| Source Model | Methods | SDv1.5 | SDv2.0 | SDv2.1 | SDv3.0 |
|---|---|---|---|---|---|
| SDv1.5-Inpainting | AdvI2I | 82.5% | 80.5% | 84.0% | 34.0% |
| | Adaptive | 78.5% | 73.5% | 77.5% | 33.0% |

Table 7: ASR of AdvI2I and AdvI2I-Adaptive training on SDv1.5-Inpainting and evaluating on other SD inpainting models regarding concept nudity.

To demonstrate that AdvI2I is not architecture-specific, we evaluate its transferability from In-structPix2Pix to SD-Turbo Image-to-Image and from SDv1.5-Inpainting to FLUX.1-dev ControlNet Inpainting-Alpha. The results are shown in Table 8.

| Source Model | Target Model | ASR |
|---|---|---|
| SDv1.5-Inpainting | FLUX.1-dev ControlNet Inpainting-Alpha | 74.0% |
| InstructPix2Pix | SD-Turbo | 35.0% |

Table 8: ASRs of AdvI2I that transfers from InstructPix2Pix to SD-Turbo Image-to-Image and from SDv1.5-Inpainting to FLUX.1-dev ControlNet Inpainting-Alpha.

## A.2 ABLATION STUDIES

| Model | Concept | Method | w/o Defense | SLD | SD-NP | GN | SC |
|---|---|---|---|---|---|---|---|
| InstructPix2Pix | Nudity | W/o Generation | 18.5% | 16.0% | 17.5% | 18.5% | 11.0% |
| | | AdvI2I (ours) | **81.5%** | **78.0%** | **79.5%** | 64.5% | 18.0% |
| | | AdvI2I-Adaptive (ours) | 78.0% | 72.5% | 74.5% | **73.0%** | **70.5%** |
| | Violence | W/o Generation | 18.0% | 14.5% | 15.5% | 17.5% | 12.0% |
| | | AdvI2I (ours) | **80.0%** | **72.5%** | **74.0%** | 65.5% | 32.5% |
| | | AdvI2I-Adaptive (ours) | 75.5% | 70.5% | 73.5% | **70.0%** | **70.5%** |
| SDv1.5-Inpainting | Nudity | W/o Generation | 55.0% | 53.5% | 54.0% | 53.5% | 3.5% |
| | | AdvI2I (ours) | **82.5%** | **78.5%** | **80.0%** | 70.0% | 10.5% |
| | | AdvI2I-Adaptive (ours) | 78.5% | 75.0% | 75.5% | **72.5%** | **72.0%** |
| | Violence | W/o Generation | 52.5% | 49.0% | 49.5% | 49.0% | 31.5% |
| | | AdvI2I (ours) | **81.0%** | **75.0%** | **78.5%** | 66.5% | 31.5% |
| | | AdvI2I-Adaptive (ours) | 76.5% | 72.5% | 73.0% | **69.5%** | **71.5%** |

Table 9: The ASR of "W/o Generation" against different defense methods on the InstructPix2Pix diffusion model.

| Method | $\alpha$ | w/o Defense | SLD | SD-NP | GN | SC |
|--------|----------|-------------|-----|-------|-----|-----|
| AdvI2I | 2.2 | 80.5% | 73.5% | 76.5% | 64.5% | **20.0**% |
|        | 2.5 | 81.5% | **78.0**% | **79.5**% | 64.5% | 18.0% |
|        | 2.8 | **82.5**% | 68.0% | 73.0% | **65.5**% | 17.5% |
| Adaptive | 2.2 | 75.5% | 60.5% | 62.5% | 71.5% | 70.0% |
|          | 2.5 | **78.5**% | **75.0**% | **75.5**% | 70.5% | **72.0**% |
|          | 2.8 | 76.5% | 72.5% | 74.0% | **73.5**% | 68.0% |

Table 10: Comparison of different $\alpha$ scales with various defense methods.

**Performance of AdvI2I w/o Using Generator.** We evaluate the performance of the method "W/o Generation" for the ablation study, which directly optimizes adversarial perturbations on the image. As shown in Table 9, W/o Generation perform much worse than AdvI2I, since it lacks the ability to generalize adversarial noise effectively.

**Varying scale of concept $\alpha$.** The influence of the concept strength parameter $\alpha$ on attack effectiveness, as shown in Table 10, underscores the importance of carefully tuning this parameter. As $\alpha$ increases, the attack becomes more aggressive, reaching a peak ASR at 82.5% without defense. However, even with stronger adversarial concepts, defenses like SC and SLD manage to reduce the ASR to moderate levels, indicating their capacity to counterbalance the attack's growing intensity. This suggests that while higher $\alpha$ values amplify the attack's potential, they also expose it to more effective defensive countermeasures. The adaptive version of AdvI2I demonstrates that balancing attack strength and defense resilience is critical, as it maintains higher ASRs despite the defenses.

### A.3 CONFIGURATION OF THE SAFE LATENT DIFFUSION (SLD)

We observe that even the "Medium" strength setting of SLD can substantially degrade the quality of images generated during benign image editing tasks with I2I diffusion models. To address this issue and enhance compatibility with I2I diffusion models, we adjust the SLD configuration accordingly. Specifically, we set the guidance scale to 1000, the warmup step to 7, the threshold to 0.01, the momentum scale to 0.3, and $\beta$ to 0.4.

### A.4 RESULTS ON THE SDv2.1-INPAINTING MODEL

We evaluate AdvI2I on the SDv2.1-Inpainting model. As shown in Table 11, it achieves an ASR of 78.5% under the nudity concept, demonstrating that AdvI2I can generalize to state-of-the-art diffusion models.

| Concept | Method | w/o Defense | SLD | SD-NP | GN | SC |
|---------|--------|-------------|-----|-------|-----|-----|
| Nudity | Attack VAE | 35.5% | 32.5% | 35.0% | 32.5% | 7.0% |
|        | MMA | 38.0% | 32.5% | 36.5% | 23.5% | 37.0% |
|        | **AdvI2I (ours)** | **78.5**% | **73.0**% | **75.0**% | **64.5**% | 10.5% |

Table 11: The ASR of different attack strategies against different defense methods on the SDv2.1-Inpaining diffusion model.

| Source Safety Checker | Target Safety Checke | ASR |
|-----------------------|----------------------|-----|
| ViT-L/14-based | ViT-L/14-based | 72.0% |
|                | ViT-B/32-based | 66.5% |

Table 12: The ASR of AdvI2I-Adaptive transferred to different safety checkers.

## A.5 THE TRANSDERABILITY OF ADVI2I-ADAPTIVE ON DIFFERENET SAFETY CHECKERS

In our work, we consider a ViT-L/14-based NSFW-detector as the safety checker. We also evaluate the transferability of AdvI2I-Adaptive on SDv1.5-Inpainting to a ViT-B/32-based NSFW-detector and observe that it still achieves a high ASR, as shown in Table 12.

## A.6 THE EVALUATION OF THE IMAGE QUALITY

We provide a comparison of the quality of attacked images using LPIPS, SSIM, PSNR, FSIM, and VIF. The results are in Table 13. The results highlight that AdvI2I performs on par with Attack VAE in terms of structural and perceptual similarity (SSIM and LPIPS) and visual feature retention (FSIM and VIF), while significantly outperforming MMA. Importantly, both AdvI2I and Attack VAE use generators to produce adversarial images, while MMA directly optimizes adversarial noise. Although MMA achieves a higher PSNR due to its direct noise optimization approach, it performs worse in metrics like VIF and SSIM. AdvI2I successfully balances adversarial effectiveness and attacked image quality across all metrics, reinforcing its stealthiness and robustness.

We include Face-Adapter (Han et al., 2025), a diffusion-based face swap method using SDv1.5 as the base model, as a baseline for comparison. The image quality is evaluated using multiple metrics: TOPIQ with three checkpoints trained on different datasets: flive, koniq, and spaq) (Chen et al., 2024), NIQE, PIQE, and FID. As shown in Table 14, AdvI2I consistently performs competitively across various metrics. It achieves higher quality in TOPIQ-koniq and TOPIQ-spaq compared to Face-Adapter, while also showing significant improvements in NIQE, PIQE, and FID scores, which indicate better perceptual quality and closer alignment to real image distributions. These results demonstrate that AdvI2I effectively generates high-quality adversarial images while maintaining its primary objective of exposing vulnerabilities in I2I models.

| Method | LPIPS↓ | SSIM↑ | PSNR↑ | FSIM↑ | VIF↑ | ASR(%)↑ |
|---|---|---|---|---|---|---|
| Attack VAE | 0.31 | 0.89 | 18.80 | 0.96 | 0.73 | 41.5 |
| MMA | 0.32 | 0.63 | 23.19 | 0.94 | 0.35 | 42.0 |
| **AdvI2I (ours)** | 0.31 | 0.88 | 18.79 | 0.96 | 0.72 | 82.5 |

Table 13: Comparison of structural and perceptual similarity metrics for attacked images across different methods.

| Method | TOPIQ-koniq↑ | TOPIQ-flive↑ | TOPIQ-spaq↑ | NIQE↓ | PIQE↓ | FID↓ |
|---|---|---|---|---|---|---|
| Face-Adapter | 0.43 | 0.83 | 0.50 | 6.36 | 62.60 | 104.63 |
| **AdvI2I (ours)** | 0.58 | 0.78 | 0.67 | 3.76 | 38.72 | 85.60 |

Table 14: Comparison of image quality metrics between AdvI2I and Face-Adapter across various metrics.

