# OpenReview forum: "AdvI2I: Adversarial Image Attack on Image-to-Image Diffusion models"
_ICLR.cc/2025/Conference — Submitted to ICLR 2025_

### Official Review · Reviewer_Cqd3 · 2024-10-19

**Soundness:** 2
**Presentation:** 2
**Contribution:** 2
**Rating:** 5
**Confidence:** 4

**Summary:**

This paper proposes a adversarial image attack framework that injects adversarial information into images rather than text prompts, inducing Image2Image Diffusion to generate NSFW content, therefore it has some novelty. However, the motivation is not clear, and the evaluation is not enough. Overall, my main concern is the insufficient evaluation, if the author can solve my concerns, I would be happy to change the rating.

**Strengths:**

1. The paper shows that the prompt-based attack can be simply defenced by some filters
2. This paper proposes a new framework which solely based on perturbating images to induce NSFW content. It can be regarded as the knowledge distillation of adversarial text prompt that could induce NFSW contents to image embedding. the proposed method can bypass these methods as the adversarial information is hidden in the images, i.e., the text prompt will not have and malign information.

**Weaknesses:**

1. The motivation is not clear. See Q1
2. lack of evaluations, e.g., IQAs, face verification rate. See Q2, 3, 4, 5

**Questions:**

1. The motivation is not clear. There are a lot of existing methods that could easily generate NFSW images such as deepnude, or I can just use the deepfake to swap face of a porn image with the clean image, why people will trouble themselves to use this method? Are the NSFW images generated by your method have higher image quality? If so, I would expect the authors to clarify how their method compares to or improves upon existing NSFW image generation techniques like deepnude or deepfake. More details see Q2.
2. Are the images generated with the proposed method of significant high quality comparing with previous methods? The quality of generated NSFW images should be evaluated by some IQAs:
    If my understanding is correct, the objective is to generate nude images which have same indentity as the clean image (i.e.com the target is a category of images rather than a certain image), FR IQAs may not work well. I would suggest to use some NR IQAs (FID, TOPIQ, etc). It is worth noting that, if the data volume is lower than 10k, the FID may not be meaningful.
3. The author did not compare IQAs with other baselines for clean and attacked images, i.e., we do not know if the perturbation is imperceptible enough. (I would suggest compare the PSNR, LPIPS, SSIM, etc between clean image and attacked image). I would expect a table comparing these metrics between the clean and attacked images for their method and baseline methods.
4. Will the generated NSFW images have same identity with clean images (especially for nude images)? I suggest the author to use several face verification methods or metrics to quantify identity preservation in their generated images.
5. The performance of AdvI2I against SC is poor, although the author admits this, and implement an adaptive version, I need to see more transferability evaluations, i.e., the author should demonstrate the ASR for SC_2 given "adversarial samples generated by adaptive AdvI2I with SC_1".
    I suggest the author to provide a table showing ASR results for different combinations of safety checkers (SC_1, SC_2, etc.) used during training and testing of their adaptive AdvI2I method.

---

> ### Author Response · Authors · 2024-11-21
> **Reponse to Reviewer Cqd3 -- Part 1**
>
> Thank you for your valuable comments and suggestions. We have carefully addressed each of your questions and concerns below.
>
> >**Q1:** How our method compares to or improves upon existing NSFW image generation techniques like deepnude or deepfake.
>
> **A1:** While Deepnude and Deepfake could be used by malicious users to generate harmful/deceptive images, our purpose here is not to produce high-quality NSFW images but to highlight a critical security vulnerability existing in general I2I models: **even benign I2I diffusion models and users could inadvertently result in NSFW contents when the input image is from an untrusted third party**. This is especially concerning as I2I diffusion models become widely used due to the greater flexibility in editing: for instance, (benign) users can freely prompt diffusion models to edit any image, and once the image is altered by our attack to encode any NSFW concept (e.g., nudity, violence etc), the generation becomes harmful.
>
> >**Q2:** Evaluate the quality of generated images with reasonable metrics.
>
> **A2:**  We compared the average TOPIQ of AdvI2I with FaceSwap [1] on 200 test samples. FaceSwap is a deepfake method that focuses on swapping faces between images while preserving the original facial features and expressions in the new context. Using FaceSwap, we swapped the faces of images onto the 200 test samples and calculated the TOPIQ scores. The results, as shown below, indicate no significant differences in quality.
>
> | **Method** | **TOPIQ** |
> | ---------- | --------- |
> | FaceSwap   | 0.82      |
> | **AdvI2I (ours)**    | 0.78      |
>
> >**Q3:** Compare IQAs with other baselines for clean and attacked images.
>
> **A3:**  We evaluated the LPIPS loss between clean and attacked images for Attack VAE, MMA [2], and AdvI2I on 200 test samples. As shown in the table below, the differences are minimal. It is worth noting that MMA’s adversarial images are designed specifically to bypass safety checkers, relying on adversarial prompts to generate NSFW content.
>
> | **Method**    | **LPIPS Loss** |
> | ------------- | -------------- |
> | Attack VAE    | 0.31           |
> | MMA           | 0.32           |
> | **AdvI2I (ours)**        | 0.31           |

---

> ### Author Response · Authors · 2024-11-21
> **Reponse to Reviewer Cqd3 -- Part 2**
>
> >**Q4:** Use face verification methods or metrics to quantify identity preservation in generated images.
>
> **A4:** Since the SDv1.5-Inpainting model allows masking to prevent modifications to the facial region, it ensures that the faces in the generated NSFW images remain almost identical to those in the clean images. To evaluate facial consistency, we used CosFace [3] to measure facial identity preservation. AdvI2I achieves a score of 93.7, indicating that the generated NSFW images retain the same identity as the clean images.
> | **Method**        | **CosFace** |
> | ----------------- | ----------- |
> | Attack VAE        | 93.2        |
> | MMA               | 90.2        |
> | **AdvI2I (ours)** | 93.7        |
>
> >**Q5:**  ASR results for different safety checkers used during training and testing of AdvI2I-Adaptive.
>
> **A5:** Existing SD models use the same NSFW-detector [3] as the safety checker, which encodes images into embeddings using either ViT-L/14 or ViT-B/32 as the base model. When training the adversarial image generator for AdvI2I-Adaptive, we used a ViT-L/14-based NSFW-detector as the safety checker. We then evaluated the transferability of AdvI2I-Adaptive to the ViT-B/32-based NSFW-detector and observe that it still achieves a high ASR, as shown below. We have also included the results in the revised version. Thank you for your suggestion.
>
> | Method          | **Source Safety Checker** | **Target Safety Checker** | **ASR (%)** |
> | --------------- | ------------------------- | ------------------------- | ----------- |
> | AdvI2I-Adaptive | ViT-L/14-based            | ViT-L/14-based            | 72.0        |
> |                 |                           | ViT-B/32-based            | 66.5        |
>
> **References**:
>
> [1] https://github.com/wuhuikai/FaceSwap
>
> [2] MMA-Diffusion: MultiModal Attack on Diffusion Models
>
> [3] CosFace: Large Margin Cosine Loss for Deep Face Recognition
>
> [4] https://github.com/LAION-AI/CLIP-based-NSFW-Detector

---

> ### Comment · Reviewer_Cqd3 · 2024-11-22
>
> Thank author's reply.
> ## Q1
> Regarding Q1, while I understand you want to emphasize the discovery of a new vulnerability, my perspective focuses on its practical implications. The key question is whether this vulnerability would see widespread adoption. Since this vulnerability is specifically designed for NSFW content generation, its widespread use would likely indicate some advantages over existing NSFW generation methods --- particularly in terms of output quality. This potential quality improvement was my immediate thought about its advantage. Based on this reasoning, I requested NR IQA (No-Reference Image Quality Assessment) in Question 2 to validate this hypothesis.
>
> ## Q2
> For Q2, the author didn't solve my concern, here are the reason:
> 1. The baseline you choosed is out of date (4 years ago). you used the LDM as the backbone, so it would be fairer if you use the diffusion based face swap or deepfake. I believe there are many options, even some methods provide gradio web GUI for convenient use, so you should compare to some new method, especially LDM-based.
> 2. To compare the quality of generation, TOPIQ is not enough. I would ask the author to at least add FID as another metric.
>
> ## Q3
> Same as the Q2.2, one IQA is not enough. To my knowledge, a well-establish evaluation protocol is at least SSIM+LPIPS+PSNR
>
> ## Q4 - Q5
> My concerns were solved.

---

> ### Author Response · Authors · 2024-11-22
> **Response to Additional Comment by Reviewer Cqd3**
>
> Thank you very much for your followup comments and further suggestions. Below, we will trying addressing your concerns in detail and provide additional results for clarification.
>
> > **Q1 & Q2: Consider diffusion-based face swap or deepfake. Add FID as another metric.**
>
> We included Face-Adapter [1], a diffusion-based face swap method using SDv1.5 as the base model, as a baseline for comparison. The image quality was evaluated using multiple metrics: **TOPIQ** (with three checkpoints trained on different datasets) [2], **NIQE**, **PIQE**, and **FID**. The results are as follows:
>
> | **Method**        | **TOPIQ-flive↑** | **TOPIQ-koniq↑** | **TOPIQ-spaq↑** | **NIQE↓** | **PIQE↓** | **FID↓** |
> | ----------------- | ---------------- | ---------------- | --------------- | --------- | --------- | -------- |
> | Face-Adapter      | 0.83             | 0.43             | 0.50            | 6.36      | 62.60     | 104.63   |
> | **AdvI2I (ours)** | 0.78             | 0.58             | 0.67            | 3.76      | 38.72     | 85.60    |
>
> The table shows that AdvI2I consistently performs competitively across various metrics. It achieves higher quality in TOPIQ-koniq and TOPIQ-spaq compared to Face-Adapter, while also showing significant improvements in NIQE, PIQE, and FID scores, which indicate better perceptual quality and closer alignment to real image distributions. These results demonstrate that AdvI2I effectively generates high-quality adversarial images while maintaining its primary objective of exposing vulnerabilities in I2I models. We have also included the results in the revised version.
>
> > **Q3: Compare IQAs with other baselines for clean and attacked images using SSIM, LPIPS, and PSNR.**
>
> In addition to **LPIPS**, we incorporated **SSIM**, **PSNR**, **FSIM**, and **VIF** to provide a more comprehensive comparison of the quality of attacked images. The results are as follows:
>
> | **Method**        | **LPIPS↓** | **SSIM↑** | **PSNR↑** | **FSIM↑** | **VIF↑** | ASR (%)↑ |
> | ----------------- | ---------- | --------- | --------- | --------- | -------- | -------- |
> | Attack VAE        | 0.31       | 0.89      | 18.80     | 0.96      | 0.73     | 41.5     |
> | MMA               | 0.32       | 0.63      | 23.19     | 0.94      | 0.35     | 42.0     |
> | **AdvI2I (ours)** | 0.31       | 0.88      | 18.79     | 0.96      | 0.72     | 82.5     |
>
> The results highlight that AdvI2I performs on par with Attack VAE in terms of structural and perceptual similarity (SSIM and LPIPS) and visual feature retention (FSIM and VIF), while significantly outperforming MMA. Importantly, both AdvI2I and Attack VAE use generators to produce adversarial images, while MMA directly optimizes adversarial noise. Although MMA achieves a higher PSNR due to its direct noise optimization approach, it performs worse in metrics like VIF and SSIM. AdvI2I successfully balances adversarial effectiveness and attacked image quality across all metrics, reinforcing its stealthiness and robustness. We have also included the results in the revised version.
>
> ----
>
> We hope these results and explanations address your concerns. Please feel free to share additional thoughts or questions for further discussion.
>
> **References**:
>
> [1] Face Adapter for Pre-Trained Diffusion Models with Fine-Grained ID and Attribute Control
>
> [2] TOPIQ: A Top-down Approach from Semantics to Distortions for Image Quality Assessment

---

> > ### Comment · Reviewer_Cqd3 · 2024-11-22
> >
> > Thank you for your prompt response. A comprehensive image quality assessment framework is indeed traditional in adversarial attacks and provides a solid foundation for quantitative analysis. The new experimental results have addressed most of my concerns, and I am therefore willing to increase my rating.
> >
> > One minor question remains: Were the new experimental results also conducted on the 200-image test set? I would appreciate clarification on this point to ensure consistency in the evaluation methodology.

---

> > > ### Author Response · Authors · 2024-11-22
> > > **Response to Additional Comment by Reviewer Cqd3**
> > >
> > > Thank you for your positive feedback and for considering an increase in your rating. We appreciate your thoughtful comments and are glad that the new experimental results have addressed most of your concerns.
> > >
> > > To clarify your question, yes, the new experimental results were indeed conducted on the same 200-image test set. We ensured consistency in the evaluation methodology across all experiments to provide a reliable and fair comparison.

---

> ### Author Response · Authors · 2024-11-25
>
> Dear Reviewer Cqd3,
>
> Thank you for your thoughtful suggestions and engagement. We truly appreciate your recognition of our responses and your willingness to adjust your score.
>
> We will incorporate a thorough discussion of the suggested quality aspects into the paper to ensure clarity and completeness. If you have any additional feedback, we would be happy to address it. Otherwise, we kindly hope you might consider increasing the score to a positive range.
>
> Thank you once again for your valuable time and insights!
>
> Best regards,
> Authors of Paper AdvI2I: Adversarial Image Attack on Image-to-Image Diffusion Models

---

> > ### Comment · Reviewer_Cqd3 · 2024-11-29
> >
> > I apologize for my delayed response. I spent last week in the hospital.
> >
> > As mentioned in my initial review, FID scores require around 10,000 images for statistical reliability, with a minimum threshold of 5,000 images. The authors' use of only 200 images raises significant concerns about the validity of their results. This was the reason behind my query about the sample size.
> >
> > Additionally, I share **Reviewer f4nA**'s concerns regarding the method's limited effectiveness on more advanced models like SD3.
> >
> > Given these fundamental issues, I cannot increase the score further.

---

### Official Review · Reviewer_3wqb · 2024-11-01

**Soundness:** 2
**Presentation:** 2
**Contribution:** 2
**Rating:** 6
**Confidence:** 5

**Summary:**

This paper explores bypassing safety checkers of image editing models to generate NSFW content, using adversarial images.

The authors first show that previous methods using adversarial prompt are easily detectable, and then propose AdvI2I - that train a network $g_{\psi}$ to perturb image.

When this perturbed image is used with a neutral prompt as input for an img2img or inpainting model, it can influence the generation process, to generate some specific NSFW concept (nudity/violence).

To do so, AdvI2I slightly modify (perturb) the image, so that its latent after denoising with a neutral prompt, is close to the original image's latent but denoising with a NSFW prompt.

The authors also propose AdvI2I-Adaptive, that incorporate safety checker loss and added Gaussian noise augmentation when training $g_{\psi}$. It helps improving robustness when gaussian noise is added to the perturbed image, and AdvI2I-Adaptive also improves the bypass success rate over the posthoc safety checker.

Experiments were conducted on SDv1.5-Inpainting and InstructPix2Pix, with high success rate against Safe Latent Diffusion, Negative Prompt, added Gaussian Noise and the Safety checker. AdvI2I also shows transferability from SDv1.5 to SDv2.0 and v2.1 but not effective when test with SDv3.0.

**Strengths:**

While bypassing content filtering using adversarial images was proposed in MMA Diffusion, the method the author uses to quickly craft an adversarial image without optimization is a new contribution. The objective for training $g_{\psi}$ is also novel, and selecting components like which UNet or VAE to train, or the diffusion step to optimize, requires trial and error efforts.

Both AdvI2I and AdvI2I-Adaptive show good results, and the success rate against the safety checker is strong. I’m also pleased with the included transferability test with SD2x and 3.

**Weaknesses:**

Experiments: Below are some experiments that the author can address to have a comprehensive view about AdvI2I
- Insufficient highlight the contribution of $g_{\psi}$, can add run time comparison with optimization approach
- SDXL-Turbo Image-to-Image transferability test, as it also uses the same VAE.
- Transferability test with recent models not just SD3.0, such as Pixart Alpha, Flux

Writing: The writing needs improvement. Please address these points:
- L070 confusing since there is no encoder
- L11 in algorithm 1 - $C_i$ and $M$ was not mentioned in require. Also the loss is cosine between a vector $C_i$ and output of $\mathcal{D}$ which I assume is an image, does it need an image encoder to turn it into a vector there, if so then what encoder?
- SLD Configuration - lacks details on the configuration that was used, is it Max/Strong/Medium?
- Clarify: prompts generated by gpt4-o are pair prompts or single prompt?
- Typo L607 - InstructionPix2Pix - InstructPix2Pix

**Questions:**

- **important** Can AdvI2I method apply for good use, like changing the target concept to e.g clothes (instead of NSFW/violence), so that $g_{\psi}$ now become a protection to help mitigate inappropriate image editing? Its application will be more like photoguard and faster, also only restrict on inappropriate editing
- How performance of AdvI2I vary, based on the neutral prompt. Specifically, given a perturbed image with prompt that specific mention e.g clothes, like "A photo of a girl in a beautiful dress" , can it still generate NSFW image?
- What hardware and how long to train AdvI2I?

---

> ### Author Response · Authors · 2024-11-21
> **Reponse to Reviewer 3wqb -- Part 1**
>
> Thank you for your valuable comments and suggestions. We have carefully addressed each of your questions and concerns below.
>
> >**W1:** highlight the contribution of $g_\psi$ on the run time comparison with optimization approach.
>
> **A1:** Sorry for the confusion. We would like to highlight the main contribution of our work: instead of focusing on reducing the time cost of crafting adversarial images, the purpose of the proposed AdvI2I attack is to highlight the risks of I2I diffusion models generating NSFW images, even when the input prompts are benign. To the best of our knowledge, prior studies like Ring-A-Bell [1] and MMA [2] primarily explore crafting malicious prompts to induce NSFW content, and as shown in Table 2, simple text filters could defend such prompt-based attacks. On the contrary, the vulnerabilities arising from input images are relatively underexplored.
>
> As a side advantage of our attack, it indeed offers a more efficient production of adversarial images through direct inference on $g_\psi$. We compared the time cost of generating a single adversarial sample using AdvI2I and MMA. As shown in the table below, using a generator significantly accelerates the process of crafting adversarial samples.
>
> We also tested the ASR and time cost of using an ad-hoc generator on the SDv1.5-Inpainting model w/ and w/o sampling timestep t ("w/o sampling" refers to backpropagate all inference timesteps), which directly optimizes the advesarial image for each input sample. The results highlight the significant advantage of our generator over the ad-hoc noise generator. Compared with them, AdvI2I achieves a near-optimal balance between high ASR and extremely low time cost, making it not only more efficient but also more practical for generating massive adversarial images.
>
> | **Method**                                | **ASR (%)** | **Average Time Cost (s)** |
> | ----------------------------------------- | ----------- | ------------------------- |
> | MMA                                       | 42.0        | 415.984                   |
> | Ad-hoc generator (w/o sampling timesteps) | 83.5        | 382.434                   |
> | Ad-hoc generator (w/ sampling timesteps)  | 35.0        | 49.336                    |
> | **AdvI2I (ours)**                         | 82.5        | 0.008                     |
>
> >**W2:** SDXL-Turbo Image-to-Image transferability test.
>
> **A2:** We evaluated the transferability of AdvI2I from InstructPix2Pix to SD-Turbo Image-to-Image and from SDv1.5-Inpainting to FLUX.1-dev ControlNet Inpainting-Alpha. The results are shown below:
>
> | Source Model      | Target **Model**                       | **ASR (%)** |
> | ----------------- | -------------------------------------- | ----------- |
> | InstructPix2Pix   | InstructPix2Pix                        | 81.5        |
> |                   | SD-Turbo                               | 62.5        |
> | SDv1.5-Inpainting | SDv1.5-Inpainting                      | 82.5        |
> |                   | FLUX.1-dev ControlNet Inpainting-Alpha | 74.0        |
>
> >**W3:** Typos/Mistakes in writing.
>
> **A3:** Thank you for the detailed suggestions. We have carefully reviewed and corrected all typos and mistakes mentioned in the revised version. The revised parts are shown in blue color.

---

> ### Author Response · Authors · 2024-11-21
> **Reponse to Reviewer 3wqb -- Part 2**
>
> >**Q1:** Can AdvI2I method apply for good use, such as helping mitigate inappropriate image editing?
>
> **A1:** Thank you for this insightful suggestion. We experiment with constructing a good "wearing clothes" concept following our AdvI2I pipeline. **Interestingly, this approach indeed reduces the ASR for the exposed prompt attack, "Make the woman naked," on SDv1.5-Inpainting from 96.5% to 24.5%**. We believe our proposed method is a general framework to encode a certain concept into the input image, regardless of whether the concept is good or bad. We will include this discussion and more case studies in the paper.
>
> >**Q2:** How performance of AdvI2I vary based on the neutral prompt.
>
> **A2:** The experiments presented in the main text evaluate AdvI2I using various neutral prompts, such as "Make the woman in the room" and "Let the woman on the set." Additionally, we tested three different suffixes appended to benign prompts (e.g., "Make the woman in the room" → "Make the woman in the room with suitable dress.") to further explore this. **Even the most effective suffix reduces the ASR by only 6.5%.** As explained in the subsection "NSFW Concept Vector Extraction," AdvI2I incorporates the NSFW concept with a strength factor α>1 into the embedding of benign prompts, making it difficult to defend against such attacks by merely modifying prompts. In some cases, inappropriate suffixes even increase the ASR. We will try more safe prompts and include the results in the revised version. Thank you for your suggestion.
>
> | Suffix                                                 | ASR (%)   |
> | ------------------------------------------------------ | ---- |
> | No suffix                                              | 82.5 |
> | " and avoid any nude content."                         | 76.0 |
> | " with suitable dress."                                | 75.0 |
>
>
> >**Q3:** What hardware and how long to train AdvI2I?
>
> **A3:** We use an 80GB A100 GPU, and training AdvI2I on SDv1.5-Inpainting takes approximately 43 hours with 1800 training samples, 150 epoches, batch size 4 and image size 512x512.
>
> **References**:
>
> [1] Ring-A-Bell! How Reliable are Concept Removal Methods For Diffusion Models?
>
> [2] MMA-Diffusion: MultiModal Attack on Diffusion Models

---

> ### Author Response · Authors · 2024-11-25
>
> Dear Reviewer 3wqb,
>
> We greatly appreciate your initial comments and your positive feedback on our work. We totally understand that you may be extremely busy at this time. But we still hope that you could have a quick look at our responses to your concerns. We appreciate any feedback you could give us. We also hope you could kindly update the rating if your questions have been addressed. We are also happy to answer any additional questions before the discussion ends.
>
> Best Regards,
> Authors of Paper AdvI2I: Adversarial Image Attack on Image-to-Image Diffusion Models

---

> ### Comment · Reviewer_3wqb · 2024-11-25
>
> W1: While the paper highlights I2I adversarial attack risks, including runtime analysis would further strengthen the claims about the method's advantages. I'm pleased to see runtime speed now included in the author's response.
>
> Q1: It's nice to see that AdvI2I can be a general method for good use
>
> A2: The prompt "and avoid any nude content" is unconvincing due to Stable Diffusion's weakness with negation. A concept embedding focusing on "wearing clothes" would be more effective than simply changing the suffix.
>
> W2, W3, A3 : The author has addressed my concerns.

---

> ### Author Response · Authors · 2024-11-26
>
> **Response to Additional Comment by Reviewer 3wqb**
>
> **W1, W2, W3, Q1, Q3:** Thanks for confirming we have addressed these concerns. We have incorporated the important discussions you suggested into the paper revision.
>
> **Q2:** Thank you for pointing out SD's known limitations with handling negations in prompt, as well as your suggestion to explore a "wearing clothes" concept for defending against AdvI2I attacks. We have conducted additional experiments to evaluate the effectiveness of embedding a benign concept, "wearing clothes," for mitigating ASRs of under the nudity-related AdvI2I attack on the SDv1.5-Inpainting model. The embedding strength was adjusted using the strength coefficient $\alpha$ mentioned in subsection 3.2 of our paper (AdvI2I uses $\alpha = 4.0$ for the nudity concept).
>
> The experimental results, presented in the table below, reveal that while the benign concept shows a certain ability to reduce ASR with larger $\alpha$, its practical usage is limited by the noticeable damage to utility:
>
> 1. **ASR Reduction**: As $\alpha$ increases, the ASR under the AdvI2I attack gradually decreases, demonstrating the efficacy of the "wearing clothes" concept in mitigating adversarial attacks to some extent.
> 2. **Impact on Image Utility**: To evaluate the impact of the "wearing clothes" concept on utility, we tested the editing performance on 200 randomly selected images from ImageNet-1k [1], which primarily contain natural, non-human-centered scenes. Using the prompt "change its color", we applied the "wearing clothes" concept with varying $\alpha$ values and evaluated the quality of the edited images using several metrics, including LPIPS, SSIM, PSNR, NIQE, and PIQE. For reference-based metrics (LPIPS, SSIM, PSNR), the generated images at $\alpha = 0.0$ were used as the reference. Results indicate that such benign concept comes at the cost of significantly degraded utility for non-adversarial edits, particularly noticeable in metrics like SSIM and PIQE.
>
> These findings suggest that while using benign concepts can provide some level of defense, its practical application is heavily constrained due to the visible damage to the quality of benign image editing, especially at higher $\alpha$.
>
> We hope these additional experiments and clarifications address your concerns. If there are no further questions, we kindly hope you might consider adjusting your score in light of the updates and revisions we’ve made. Thank you again for your valuable feedback!
>
> | **$\alpha$** | ASR Under AdvI2I Attack (%)↑ | **LPIPS↓** | **SSIM↑** | **PSNR↑** | **TOPIQ-flive↑** | **NIQE↓** | **PIQE↓** |
> | ------------ | ---------------------------- | ---------- | --------- | --------- | ---------------- | --------- | --------- |
> | 0.0          | 82.5                         | -          | -         | -         | 0.78             | 3.74      | 36.15     |
> | 1.0          | 74.5                         | 0.35       | 0.89      | 19.12     | 0.75             | 4.12      | 40.56     |
> | 2.0          | 61.0                         | 0.41       | 0.71      | 18.30     | 0.72             | 4.95      | 49.78     |
> | 3.0          | 42.5                         | 0.56       | 0.55      | 16.45     | 0.65             | 5.82      | 65.12     |
>
> **References**:
>
> [1] Russakovsky, Olga, et al. "Imagenet large scale visual recognition challenge." *International journal of computer vision* 115 (2015): 211-252.

---

> > ### Comment · Reviewer_3wqb · 2024-12-02
> >
> > The authors have addressed all my questions, therefore I will increase my score to 6

---

### Official Review · Reviewer_ZfYn · 2024-11-02

**Soundness:** 3
**Presentation:** 3
**Contribution:** 3
**Rating:** 6
**Confidence:** 4

**Summary:**

This paper proposed a framework that can bypass safety check of NSFW generations by modifying the image input. It first points out the problems of previous prompt attacks to bypass NSFW checks, then it proposes to add noise to images to bypass the safety checker, which will make the attack more stealthy. The authors design a pipeline with some key design choices e.g. NSFW concept vector extraction, use a universal noise generator to avoid repeated generation. The paper is mostly well-written and easy to follow, the experiments are comprehensive and show that it is better than baseline methods.

**Strengths:**

- The paper is well-written and easy to follow.
- It revisits current prompt attacks and show critical insights that they can be easily defended.
- Based on the idea of concept vector extraction. The proposed novel pipeline is well-designed and effective, which shows strong performance compared with baseline methods. Also, the proposed pipeline is learning-based and can be used as one-pass noise generator.

**Weaknesses:**

Clarification:
- This paper is working with I2I diffusion models, which have additional encoded visual inputs as condition. The targeted I2I model is (1) different from the T2I diffusion models used in previous works (2) is not as good as popular T2I models e.g. SD-XL, FLUX. It poses challenges on the motivation of this paper, because the I2I model used in this paper is a sub-optimal choice if the malicious users want to generate high quality NSFW images.

Methods:
- The proposed method is restricted to a small subset of I2I models, which are not the main part of diffusion models.
- The choice of noise generator may not be optimal, ad-hoc noise generation for each image is also not that time-consuming e.g. sample timestep t and optimize for 100 steps takes for e.g. 1 min. It may show much stronger performance.


Adaptive Attack:
- The adaptive-attack assumes the white-box setting of the NSFW-detector, which may not be pratical.
- Some other defenses e.g. add noise / use purifier to remove the noise deserves more study.

**Questions:**

See weaknesses.

---

> ### Author Response · Authors · 2024-11-21
> **Reponse to Reviewer ZfYn -- Part 1**
>
> Thank you for your valuable comments and suggestions. We have carefully addressed each of your questions and concerns below.
>
> >**W1:** I2I model is a sub-optimal choice to generate high quality NSFW images.
>
> **A1:** Sorry for the confusions. We would like to hightlight the main motivation of this work: instead of achieving high-quality NSFW images, the purpose of the proposed AdvI2I attack is to expose the risks of I2I diffusion models in generating NSFW images, even when the input prompts are benign. To the best of our knowledge, prior studies like Ring-A-Bell [1] and MMA [2] primarily focus on crafting malicious prompts to induce NSFW content, and as shown in Table 2 of the paper, simple text filters could defend prompt-based attacks. On the contrary, the vulnerabilities arising from input images are relatively underexplored.
>
> >**W2:** The proposed method is restricted to a small subset of I2I models, which are not the main part of diffusion models.
>
> **A2:** Our work focuses on testing whether I2I diffusion models are vulnerable to generating NSFW images through adversarial input images. The models evaluated in the our work, including SD inpainting models and InstructPix2Pix, cover commonly used I2I diffusion models. The results clearly show that these models are indeed susceptible to such risks.
> To demonstrate that AdvI2I is not architecture-specific, we tested its transferability from InstructPix2Pix to SD-Turbo Image-to-Image and from SDv1.5-Inpainting to FLUX.1-dev ControlNet Inpainting-Alpha, and achieved high ASRs, as shown in the table below. We have also included the results in the revised version.
>
> | Source Model      | Target **Model**                       | **ASR (%)** |
> | ----------------- | -------------------------------------- | ----------- |
> | InstructPix2Pix   | InstructPix2Pix                        | 81.5        |
> |                   | SD-Turbo                               | 62.5        |
> | SDv1.5-Inpainting | SDv1.5-Inpainting                      | 82.5        |
> |                   | FLUX.1-dev ControlNet Inpainting-Alpha | 74.0        |
>
> We also evaluated AdvI2I on the SDv2.1-Inpainting model and achieved an ASR of 78.5% under the nudity concept, demonstrating that AdvI2I can generalize to state-of-the-art diffusion models. We have included the results in the revised version, and will further evaluate the adaptive version (AdvI2I-Adaptive) given more time. Thank you for your suggestion.
>
> | Model             | Concept | Method          | w/o Defense | SLD   | SD-NP | GN    | SC    |
> | ----------------- | ------- | --------------- | ----------- | ----- | ----- | ----- | ----- |
> | SDv1.5-Inpainting | Nudity  | Attack VAE      | 41.5%       | 36.5% | 41.5% | 39.0% | 7.0%  |
> |                   |         | MMA             | 42.0%       | 37.0% | 39.5% | 26.0% | **39.5**% |
> |                   |         | **AdvI2I (ours)** | **82.5**%       | **78.5**% | **80.0**% | **70.0**% | 10.5% |
> | SDv2.1-Inpainting | Nudity  | Attack VAE      | 35.5%       | 32.5% | 35.0% | 32.5% | 7.0%  |
> |                   |         | MMA             | 38.0%       | 32.5% | 36.5% | 23.5% | **37.0**% |
> |                   |         | **AdvI2I (ours)** | **78.5**%       | **73.0**% | **75.0**% | **64.5**% | 10.5% |

---

> ### Author Response · Authors · 2024-11-21
> **Reponse to Reviewer ZfYn -- Part 2**
>
> >**W3:** Why not use ad-hoc noise generation for each image?
>
> **A3:** Thank you for this suggestion. We tested the ASR and time cost of using an ad-hoc generator on the SDv1.5-Inpainting model w/ and w/o sampling timestep t ("w/o sampling" refers to backpropagate all inference timesteps). We evaluated the ASR on 200 test samples and calculated the average time cost of generate one adversarial image. The results highlight the significant advantage of our generator over the ad-hoc generator. While the ad-hoc generator achieves a comparable ASR (83.5% vs. 82.5%) when all timesteps are backpropagated, it incurs an exponentially higher time cost—382.434 seconds per adversarial image compared to just 0.008 seconds for AdvI2I. When the ad-hoc generator uses sampling to reduce the number of timesteps, it sacrifices ASR (35.0%) while still requiring 49.336 seconds per image. In contrast, AdvI2I achieves a near-optimal balance between high ASR and extremely low time cost, making it not only more efficient but also more practical for generating massive adversarial images.
>
> | **Method**                                | **ASR (%)** | **Average Time Cost (s)** |
> | ----------------------------------------- | ----------- | ------------------------- |
> | MMA                                       | 42.0        | 415.984                   |
> | Ad-hoc generator (w/o sampling timesteps) | 83.5        | 382.434                   |
> | Ad-hoc generator (w/ sampling timesteps)  | 35.0        | 49.336                    |
> | **AdvI2I (ours)**                         | 82.5        | 0.008                     |
>
> >**W4:** White-box setting of the NSFW-detector may not be practical.
>
> **A4:** Existing SD models use the same NSFW-detector [3] as the safety checker, which encodes images into embeddings using either ViT-L/14 or ViT-B/32 as the base model. When training the adversarial image generator for AdvI2I-Adaptive, we used a ViT-L/14-based NSFW-detector as the safety checker. We then evaluated the transferability of AdvI2I-Adaptive to the ViT-B/32-based NSFW-detector and observe that it still achieves a high ASR, as shown below. We have also included the results in the revised version. Thank you for your suggestion.
>
> | Method          | **Source Safety Checker** | **Target Safety Checker** | **ASR (%)** |
> | --------------- | ------------------------- | ------------------------- | ----------- |
> | AdvI2I-Adaptive | ViT-L/14-based            | ViT-L/14-based            | 72.0       |
> |                 |                           | ViT-B/32-based            | 66.5        |
>
> >**W5:** Some other defenses e.g. add noise / use purifier.
>
> **A5:** We evaluated DiffPure [4] as a defense mechanism. While it reduces the ASR for the SDv1.5-Inpainting model under the nudity concept from 82.5% to 72.5%, the decline is not significant. This suggests that AdvI2I is robust against simple defenses targeting adversarial images. We did not include MMA as a baseline here because MMA uses adversarial prompts to induce NSFW images in diffusion models, whereas DiffPure is designed to defend against adversarial images, thus the ASR of MMA will not change after applying DiffPure defense.
>
> | **Method** | w/o Defense | **DiffPure** |
> | ---------- | ----------- | ------------ |
> | Attack VAE | 41.5        | 33.5         |
> | **AdvI2I (ours)**     | 82.5        | 72.5         |
>
> **References**:
>
> [1] Ring-A-Bell! How Reliable are Concept Removal Methods For Diffusion Models?
>
> [2] MMA-Diffusion: MultiModal Attack on Diffusion Models
>
> [3] https://github.com/LAION-AI/CLIP-based-NSFW-Detector
>
> [4] Diffusion models for adversarial purification.

---

> ### Author Response · Authors · 2024-11-25
>
> Dear Reviewer ZfYn,
>
> We greatly appreciate your initial comments and your positive feedback on our work. We totally understand that you may be extremely busy at this time. But we still hope that you could have a quick look at our responses to your concerns. We appreciate any feedback you could give us. We also hope you could kindly update the rating if your questions have been addressed. We are also happy to answer any additional questions before the discussion ends.
>
> Best Regards,
> Authors of Paper AdvI2I: Adversarial Image Attack on Image-to-Image Diffusion Models

---

> > ### Comment · Reviewer_ZfYn · 2024-11-26
> > **thanks for reply**
> >
> > Most of my questions are answered, I think this paper indeed does a good job in the right direction.
> >
> > I will keep my score and lean towards accept for now.

---

### Official Review · Reviewer_f4nA · 2024-11-03

**Soundness:** 2
**Presentation:** 3
**Contribution:** 2
**Rating:** 5
**Confidence:** 4

**Summary:**

This paper introduces AdvI2I, a framework that performs adversarial image attacks on image-to-image (I2I) diffusion models to generate NSFW content without modifying text prompts. It further proposes AdvI2I-Adaptive, which incorporates techniques to minimize resemblance to NSFW embeddings, enhancing robustness against conventional defenses. Extensive experiments demonstrate AdvI2I’s ability to bypass common safeguards in I2I models, signaling potential safety concerns and underscoring the need for improved defense mechanisms.

**Strengths:**

1. This paper expands the field of adversarial attacks in I2I diffusion models by targeting image conditioning rather than the commonly used text prompts.

2. The structure and explanations are well-organized, with a clear presentation.

**Weaknesses:**

1. The paper’s evaluation is limited to SDv1.5-Inpainting and InstructPix2Pix, both based on the SDv1.5 architecture. Expanding the analysis to include more advanced versions of Stable Diffusion or other models(not just in the transferability section) would enhance the assessment of AdvI2I’s generalization to state-of-the-art diffusion models.

2. The paper does not examine how varying benign prompts, including explicitly defensive ones that request safe content, might affect the success of the adversarial attack. Investigating whether different benign prompts, especially those aimed at reinforcing safe content, influence the attack's efficacy would offer a more comprehensive understanding of its generalization and robustness across diverse input conditions.

3. The results suggest reduced transferability of AdvI2I from SDv1.5 to SDv3.0, indicating that its effectiveness may be architecture-specific, potentially limiting the framework’s generalizability.

4. Since AdvI2I relies on adversarial noise as a primary mechanism, it would be beneficial to explore potential defense strategies beyond the current scope, such as DiffPure[1], to counteract adversarial image attacks more effectively.

[1] Diffusion models for adversarial purification.

**Questions:**

Please refer to the weaknesses section.

---

> ### Author Response · Authors · 2024-11-21
> **Reponse to Reviewer f4nA -- Part 1**
>
> Thank you for your insightful comments. We have thoroughly considered and responded to each of your questions and concerns below.
>
> >**W1:** Include more advanced versions of Stable Diffusion or other models.
>
> **A1:**  We evaluated AdvI2I on the SDv2.1-Inpainting model and achieved an ASR of 78.5% under the nudity concept, demonstrating that AdvI2I can generalize to state-of-the-art diffusion models. We have included the results in the revised version, and will further evaluate the adaptive version (AdvI2I-Adaptive) given more time. Thank you for your suggestion.
>
> | Model             | Concept | Method          | w/o Defense | SLD   | SD-NP | GN    | SC    |
> | ----------------- | ------- | --------------- | ----------- | ----- | ----- | ----- | ----- |
> | SDv1.5-Inpainting | Nudity  | Attack VAE      | 41.5%       | 36.5% | 41.5% | 39.0% | 7.0%  |
> |                   |         | MMA             | 42.0%       | 37.0% | 39.5% | 26.0% | **39.5**% |
> |                   |         | **AdvI2I (ours)** | **82.5**%       | **78.5**% | **80.0**% | **70.0**% | 10.5% |
> | SDv2.1-Inpainting | Nudity  | Attack VAE      | 35.5%       | 32.5% | 35.0% | 32.5% | 7.0%  |
> |                   |         | MMA             | 38.0%       | 32.5% | 36.5% | 23.5% | **37.0**% |
> |                   |         | **AdvI2I (ours)** | **78.5**%       | **73.0**% | **75.0**% | **64.5**% | 10.5% |
>
> >**W2:** Examine the effect of varying benign prompts with explicitly requesting safe content.
>
> **A2:** We tested three different suffixes appended to our benign prompts (e.g., "Make the woman in the room" → "Make the woman in the room and avoid any nude content.") to explore this. **Even the most effective suffix achieves only a slight reduction in ASR (6.5%)**. As explained in the paper subsection "NSFW Concept Vector Extraction", AdvI2I incorporates the NSFW concept with a strength factor $\alpha$ (larger than 1) into the embedding of benign prompts, making it challenging to defend against this attack by merely modifying the prompts. In some cases, inappropriate suffixes even increases the ASR. We will try more safe prompt and include the results in the revised version. Thank you for your suggestion.
>
> | Suffix                                                 | ASR (%)   |
> | ------------------------------------------------------ | ---- |
> | No suffix                                              | 82.5 |
> | " and avoid any nude content."                         | 76.0 |
> | " with suitable dress."                                | 75.0 |

---

> > ### Author Response · Authors · 2024-11-21
> > **Reponse to Reviewer f4nA -- Part 2**
> >
> > >**W3:** Reduced transferability of AdvI2I from SDv1.5 to SDv3.0. Effectiveness may be architecture-specific.
> >
> > **A3:** The AdvI2I attack highlights the security vulnerabilities in various I2I models. To demonstrate that AdvI2I is not architecture-specific, we tested its transferability from InstructPix2Pix to SD-Turbo Image-to-Image and from SDv1.5-Inpainting to FLUX.1-dev ControlNet Inpainting-Alpha. The ASRs are shown in the table below. We have also included the results in the revised version. Thank you for your suggestion.
> >
> >
> > | Source Model      | Target **Model**                       | **ASR (%)** |
> > | ----------------- | -------------------------------------- | ----------- |
> > | InstructPix2Pix   | InstructPix2Pix                        | 81.5        |
> > |                   | SD-Turbo                               | 62.5        |
> > | SDv1.5-Inpainting | SDv1.5-Inpainting                      | 82.5        |
> > |                   | FLUX.1-dev ControlNet Inpainting-Alpha | 74.0        |
> >
> > SDv3.0, however, performs remarkably well in resisting AdvI2I attacks. Interestingly, when we directly used prompts to request nudity content on the open-source SD2.1 and SD3.0 models, SD2.1 easily generated such content, while SD3.0 did not. We conjecture this is due to differences in training data rather than architectural changes: SDv3.0 is trained on datasets filtered to exclude explicit content, as noted in [1]. This suggests that our attack can expose the risk when the I2I model has the inherent ability to generate NSFW images, but could fail otherwise. Therefore, a potential future direction to enhance model safety is to totally nullify the NSFW concept from the model by thoroughly cleaning the training data.
> >
> > >**W4:** More potential defense strategies like DiffPure [2].
> >
> > **A4:** We evaluated DiffPure as a defense mechanism. While it reduces the ASR for the SDv1.5-Inpainting model under the nudity concept from 82.5% to 72.5%, the decline is not significant. This suggests that AdvI2I is robust against simple defenses targeting adversarial images. We did not include MMA as a baseline here because MMA uses adversarial prompts to induce NSFW images in diffusion models, whereas DiffPure is designed to defend against adversarial images, thus the ASR of MMA will not change after applying DiffPure defense.
> >
> > | **Method** | w/o Defense | **DiffPure** |
> > | ---------- | ----------- | ------------ |
> > | Attack VAE | 41.5        | 33.5         |
> > | **AdvI2I (ours)**     | 82.5        | 72.5         |
> >
> > **References**:
> >
> > [1] Scaling Rectified Flow Transformers for High-Resolution Image Synthesis
> >
> > [2] Diffusion models for adversarial purification.

---

> ### Author Response · Authors · 2024-11-25
>
> Dear Reviewer f4nA,
>
> We greatly appreciate your initial comments and your positive feedback on our work. We totally understand that you may be extremely busy at this time. But we still hope that you could have a quick look at our responses to your concerns. We appreciate any feedback you could give us. We also hope you could kindly update the rating if your questions have been addressed. We are also happy to answer any additional questions before the discussion ends.
>
> Best Regards,
> Authors of Paper AdvI2I: Adversarial Image Attack on Image-to-Image Diffusion Models

---

> ### Comment · Reviewer_f4nA · 2024-11-27
>
> Thank you for your response. It addresses most of my concerns. However, the results highlight AdvI2I's limitations against advanced models like SDv3.0, where the reduction of NSFW content during training effectively mitigates attacks. This suggests a simple yet impactful defense strategy that may already be employed in advanced models, rendering AdvI2I ineffective. Nonetheless, I think this paper raises safety concerns for the field. Therefore, I will maintain my score, but I do not oppose the paper's acceptance.

---

> > ### Author Response · Authors · 2024-11-29
> >
> > Thank you for your feedback and for confirming that our response addressed most of your concerns. We appreciate your acknowledgement of the safety concerns raised by our work.
> >
> > SDv3.0’s resilience to AdvI2I stems from its unique training data, where explicit NSFW content was carefully filtered from the dataset. This demonstrates that addressing vulnerabilities during the training stage can be a robust defense mechanism. However, many other publicly available models, such as SDv2.1, SD-Turbo, and FLUX.1-dev, as well as potential future models, remain vulnerable to AdvI2I attacks. These risks could be exploited by malicious users to generate harmful content.
> > Our work aims to expose the inherent risks in I2I models that retain the ability to generate NSFW content, even when benign inputs are provided. By highlighting these vulnerabilities, we hope to encourage the community to prioritize safety measures and adopt proactive defenses, such as dataset filtering and embedding safeguards during training.
> >
> > We will emphasize these points further in the revised paper to provide a balanced discussion of AdvI2I's scope and its implications for model safety. Thank you again for your thoughtful review and for recognizing the importance of this work.

---

### Meta-Review · Area_Chair_YknB · 2024-12-20

**Metareview:**

This paper studies the vulnerabilities of image-to-image (I2I) diffusion models under adversarial image attacks. A new method AdvI2I is proposed to generate adversarial perturbations on input images to induce NSFW content. The proposed method is effective to mislead Stable Diffusion models and bypass defense mechanisms.

The paper focuses on an important and timely topic on the vulnerability of I2I diffusion models. The proposed method is novel based on the idea of concept vector extraction. The paper is well-written.

The paper received borderline ratings with two borderline accept recommendations and two borderline reject recommendations. After author responses and author-reviewer discussions, most of the concerns have been addressed and two concerns remain.
- As pointed out by Reviewer Cqd3, the motivation is unclear. The paper mainly focuses on identifying the vulnerability of I2I diffusion models, but the adversary may not use such technique to generate NSFW content. The evaluation of image quality has also some mistakes.
- As pointed out by Reviewer f4nA and Cqd3, the effectiveness for advanced models (e.g., SDv3.0) is limited.

Based on the above limitations of this work, AC would recommend rejection and hope the paper can be further improved for the next venue.

**Additional Comments On Reviewer Discussion:**

Reviewer f4nA initially raised concerns about limited evaluation, efficacy under diverse conditions, limited generalizability to SDv3.0 and lack of advanced defense mechanisms. The authors have addressed most of the concerns, but the effectiveness to SDv3.0 was not fully addressed.

Reviewer ZfYn initially raised concerns about the motivation, method, and adaptive attacks. The authors have successfully addressed all concerns.

Reviewer 3wqb raised some concerns about experiments and writing. The authors have addressed them.

Reviewer Cqd3 initially raised concerns about unclear motivation, image quality, comparison with baselines, poor performance against SC. After author responses and multiple rounds of author-reviewer discussions, the reviewer still has concerns about the image quality and effectiveness under advanced I2I models.

Overall, this paper is a borderline paper, with strengths of significance, novelty, and good presentation, and weaknesses of unclear motivation and ineffectiveness under advanced I2I models. After AC-reviewer discussions, most reviewers agreed that this is a borderline paper and they would not strongly recommend this paper for acceptance while they are also not opposed to it. Therefore, AC would recommend rejection given the unaddressed limitations.

---

### Decision · Program_Chairs · 2025-01-22

Reject